# Structural insights into the RNA interaction with Yam bean Mosaic virus (coat protein) from *Pachyrhizus erosus* using bioinformatics approach

Varsha Acharya[1], R. Arutselvan[1], Kalidas Pati[1]*, Ajaya Kumar Rout[2], Budheswar Dehury[2¤], V. B. S. Chauhan[1], M. Nedunchezhiyan[1]

**1** Regional Centre, ICAR-Central Tuber Crops Research Institute Bhubaneswar, Odisha, India, **2** ICAR-Central Inland Fisheries Research Institute, Barrackpore, Kolkata, West Bengal, India

¤ Current Address: Present address: ICMR-Regional Medical Research Centre, Chandrasekharpur, Bhubaneswar, Odisha, India

* kalidas9555@gmail.com

**Data Availability Statement:** All relevant data are within the manuscript and Supporting Information files, including NCBI accession numbers.

## Abstract

Plants are constantly threatened by a virus infection, i.e., Potyviruses, the second largest genus of plant viruses which results in several million-dollar losses in various essential crops globally. Yam bean (*Pachyrhizus erosus*) is considered to be one of the essential tuberous legume crops holding a great potential source of starch. Yam Bean Mosaic Virus (YBMV) of Potyvirus group belonging to the family potyviridae affects Yam bean and several angiosperms both in the tropical and sub-tropical regions causing large economical losses in crops. In this study, we attempted to understand the sequence-structure relationship and mode of RNA binding mechanism in YBMV CP using in silico integrative modeling and all-atoms molecular dynamics (MD) simulations. The assembly of coat protein (CP) subunits from YBMV and the plausible mode of RNA binding were compared with the experimental structure of CP from Watermelon mosaic virus potyvirus (5ODV). The transmembrane helix region is present in the YBMV CP sequence ranging from 76 to 91 amino acids. Like the close structural-homolog, 24 CPs monomeric sub-units formed YBMV a conserved fold. Our computational study showed that ARG[124], ARG[155] and TYR[151] orient towards the inner side of the virion, while, THR[122], GLN[125], SER[92], ASP[94] reside towards the outer side of the virion. Despite sharing very low sequence similarity with CPs from other plant viruses, the strongly conserved residues Ser, Arg, and Asp within the RNA binding pocket of YBMV CP indicate the presence of a highly conserved RNA binding site in CPs from different families. Using several bioinformatics tools and comprehensive analysis from MD simulation, our study has provided novel insights into the RNA binding mechanism in YBMV CP. Thus, we anticipate that our findings from this study will be useful for the development of new therapeutic agents against the pathogen, paving the way for researchers to better control this destructive plant virus.

**Funding:** The author(s) received no specific funding for this work.

**Competing interests:** The authors have declared that no competing interests exist

## Introduction

Yam bean (*Pachyrhizus erosus*), also known as "Jicama" belongs to the family Fabaceae is an economically high-yielding leguminous tuber crop showing the widest adaptation with an enhanced nutritional value [1]. It is cultivated broadly in Mexico, Singapore, China, the Philippines, Indonesia, Hawaii, and India. It is the native crop of Tropical America, some parts of Africa, and Asia. In India, it is cultivated in many regions of Assam, Orissa, Bihar, Tripura and West Bengal [2]. Yam bean mosaic virus was discovered for the first time in Brazil [3]. Yam bean (*Pachyrrhizus erosus*) is also one of the well-known legumes known as "Bengkuang" widely found in Indonesia. It is cultivated mainly in several regions of West and Central Java in Indonesia [4]. Infected plants typically exhibit leaf green vein banding, moulting, curling, inter-veinal mosaic, stunted development and the disease has been known to cause severe production losses in yam [5, 6]. The virus is distributed non-persistently through aphids and vegetative propagation. YMV has been identified in various Dioscoreaceae plants [7–10] and can be mechanically transferred to *Chenopodium amanticolor*, *Nicotiana megalosiphon* and *Nicotiana benthamiana* [8]. Yam bean serves a key function in delivering food and protein-rich micronutrients and protein for human use [10]. The subtribe Glycininae belongs to the genus *Pachyrhizus* which consists of an essential legume crop named soybean (Glycine max) [11, 12]. The genus involves three cultivated species *i.e.*, *Pachyrhizus tuberosus*, *Pachyrhizus erosus*, and *Pachyrhizus ahipa* produce roots with high storage capacity as well as with high proteins *i.e.* up to 12% in dry matter, iron content (up to 52 ppm) followed by zinc (up to 14 ppm) [13]. The genomic size of the yam bean is estimated to be between 572 and 597Mbp [14]. The most widely consumed yam bean species are *tuberosus* and *erosus*, which have different morphologies within the species and genus [15]. Viruses, on the other hand, are becoming a serious restriction for tuber crops and roots, particularly in the tropics [13].

Sorensen and co-authors reported that 20–40% of yield declination is due to certain viral diseases in *Pachyrhizus erosus* [13]. Despite of the negative impacts on yam bean crops, there is very little information available on viral diseases. The Yam bean (*P. erosus*) usually gets infected by bacteria like Pseudomonas [16], fungi [17, 18], insect-pests (*i.e.*, aphids, mealybug, and bruchids) [13], nematodes and viruses. It has been earlier reported that there are two major viruses destroying yam bean cultivation *i.e.*, bean common mosaic virus (BCMV) and sincama mosaic virus (SMV) [13]. Damayanti and colleagues identified a strain causing massive damage to Yam Bean (*Pachyrhizus erosus*) cultivation in different parts of Indonesia and recognized it as BCMV. Latter the virus was observed to be a distinct strain and named as BCMV-IYbn. Soon after in Peru, certain unique strains were observed that were suspected to be BCMV. Following that, a virus with a distinct strain was obtained for observation, and tests were performed to identify it [19]. Small interfering RNA (siRNA) sequencing took place and assembly was performed on a sample resulting in severe mosaic and leaf deformation [20]. Finally, after the analysis, it was proved that the virus was not BCMV and later Fuentes and co-authors in 2012 proposed the name of the new virus as Yam bean Mosaic Virus (YBMV). The YBMV CP comes under the Potyviridae family consisting of virions *i.e.*, non-enveloped as well as flexuous and filamentous. On the other hand, certain phylogenetic analyses were performed using only the CP (coat protein) of YBMV from the isolates of Vietnam [21] and Indonesia [19] and found maximum identity to YBMV over the CP encoding region which assures the presence of YBMV in the virus as identified as BCMV-IYbn previously. In Peru, Indonesia, and Vietnam, the presence of YBMV and associated viruses in a yard-long bean, yam bean, and black bean respectively suggests the presence of this virus in geographically remote areas and various crops [22]. The potyviral CP is a multifunctional protein that plays a part in almost every phase of the virus life cycle, in addition to its structural purpose of preserving the viral

DNA. Interactions between the viral RNA and CP, in particular, must be closely regulated for the RNA to be assigned to each of its roles correctly during the infection process [23].

Cucumber mosaic virus, infection causes mosaic symptoms in a wide range of agricultural plants, including cucumbers. The replacement of amino acid 129 in the coat protein of the Cucumber mosaic virus results in structural alterations in the virus [24]. The screening of potential inhibitors against the coat protein of the Apple Chlorotic Leaf Spot Virus was carried out. Apple chlorotic leaf spot virus (ACLSV) coat protein (CP) is an important latent virus on the apple plant that has been identified [25]. The coat protein (CP) of the tobacco mosaic virus (TMV) self-assembles in transgenic plants that have been deprived of viral RNA, forming aggregates that are dependent on the physical parameters of the environment [26]. A biosimulation technique was used to identify plant-based analogs that might serve as potential inhibitors against the tobacco mosaic virus [27]. siRNAs created by the plant's defensive system, the host RNA silencing mechanism, target the viral RNA or DNA and function in post-transcriptional gene silencing (PTGS) or transcriptional gene silencing (TGS) pathways, respectively, to inhibit viral replication. Host gene expression modulation by virus- and virus-derived small RNAs: new insights into the pathogenesis of viruses and viruses-derived small RNAs [28]. Because of its critical function in viral replication, the RNA-dependent RNA polymerase (RdRp) has emerged as a potential therapeutic target [29].

Previously, there was no protein-related information in the yam bean crop and thus there has been no work done on functional and structural protein characterization of YBMV globally. In 1988, Shukla and Ward investigated the structural characterization of the potyvirus (coat proteins) CP and observed that the amino acid sequences of CP are sufficient to identify and differentiate the individual potyvirus and its strains [30–32]. The coat protein is considered to be the major gene product in the virion [33]. The genome of potyvirus consists of eight different gene products of which the CP and the genome-linked protein are the solitary gene products in the virion [33].

Thus, in our study, we have taken the coat protein of YBMV for the structural and functional analysis of the virus CP. Since genomic structure differs significantly among families and the CPs display very low-level sequence similarity, it is fascinating to explore the virion architecture. Through integrative modeling, we explored the plausible organization of YBMV CP and its mode of single-stranded RNA (ssRNA) recognition followed by the all-atoms molecular dynamics simulation (MDS) for a 50 ns timeframe. We also identified a few conserved residues that aid in binding of RNA which suggests antiviral compounds could be targeted to these regions which may profound effect on the genome assembly and packaging of economically important viruses like YBMV. As a consequence of our research, we now have a better understanding of the structural basis of CP assembly in YBMV, which will help us in developing new antiviral drugs that target the conserved fold of RNA binding.

## Materials and methods

### Primary sequence analysis

The FASTA format amino acid sequence of YBMV CP was obtained from the NCBI protein database with accession number YP_004940328. The coat protein functional region/domain region was identified using the Conserved Domain Database (CDD) [34]. The phosphorylation site was identified using the NetPhos 2.0 server [35]. To gain broad chemistry of YBMV CP, a clear study on the primary sequence was carried out using the ProtParam tool [36] of Expasy Proteomic Server. The ProtParam provides information on different Physico-chemical properties of the coat protein, such as instability index, molecular weight, iso-electric point, grand average hydropathy (GRAVY), and aliphatic index.

## Molecular evolutionary analysis of YBMV CP

The molecular evolutionary analysis of YBMV, a phylogenetic tree was built using the NJ technique in MEGA version 6 software [37]. A total of twenty-six homologous coat protein sequences of potyvirus were collected using the DELTA-BLASTP search against the non-redundant (nr) database to construct the evolutionary genetic tree. Percentage of trees replicated where the accompanied taxa grouped in the bootstrap test (containing 1000 replicates) were signified next to the branches. The position correction method is used for determining the protein evolutionary distances through gap extension penalty, gap separation distance, and gap opening penalty employing Blossum weight matrix. All locations encompassing gaps and missing data were removed throughout the study. Using Clustal W software, multiple sequence alignment was conducted on fourteen coat proteins of Potyviruses [38].

## Secondary structural analysis

The PSIPRED server was used to identify the secondary structure of YBMV CP [39]. PSIPRED helps in sequence annotation by TM (transmembrane) domain mapping and secondary structure mapping [40]. This server is one of the specific secondary structure prediction methods that includes two feed-forward neural networks in which the analysis is based on the output obtained from Delta-Blastp.

## Template identification

Suitable templates were detected for the comparative modeling of YBMV CP, the target sequences of YBMV CP were investigated for similar sequences using the Blast-P *i.e.*, against the PDB database [41]. The matrix used for the template identification is the Blossum-62 matrix with a default inclusion threshold value (0.005) and E-value (10). The templates were chosen based on the sequence identity, less E-value, query coverage, and structural resolution. After the selection of an appropriate template, a target-template alignment was performed for the generation of protein structure.

## Computational modeling

Arnold and co-authors have developed the SWISS-MODEL workspace with a range of tools that permits the user to modify and validate the various modeling steps manually. The SWISS-MODEL workstation includes application software required for 3D structure in an easy-to-use web-based modeling computer unit. A 3D model for the target protein is generated based on the sequence alignment among the target and template structures. For structure creation, SWISS-MODEL was used to model the 3D proteins [42].

## Model validation

An essential web application such as PROCHECK has been deployed to explain the authenticity, stereochemical consistency, and durability of the projected 3D model [43]. PROCHECK was used for accessing the geometry of the model. Protein structure analysis *i.e* ProSA [44] was applied for calculating the energy potential of the predicted model, and the Verify3D [45] score used for the analysis to check the quality of the predicted model, It's also used to check if the atomic model is compatible with its amino acid sequence. Mol Probity [46] is a web service for structure validation providing a broad-spectrum solidly based evaluation of model quality at both the local and global levels for both nucleic acids and proteins [46]. In comparison to other approaches that are optimized to find native structures, ERRAT [47] was performed for estimating the accuracy of the non-bonded atoms of the modeled YMBV. The validation

results were interpreted, accompanied by re-validation to achieve an initial optimized model structure, and the structural ambiguities were resolved.

## Identification of RNA binding pocket

To identify the plausible mode of RNA recognition, the homo-oligomeric assembly (24-mers) [48] using monomeric sub-units of modeled CP of YBMV was built. Based on the potyvirus virion structure (PDB: 5DOV) [49], the oligomeric structure of CP sub-units was modeled and transferred the coordinates of RNA to our modeled structure using PyMOL. The electrostatic surface potential map was constructed using the APBS plug-in of PyMOL. To identify, the conserved RNA binding site which was reported in earlier studies [48]. We have employed HADDOCK *v*2. 4 web server to predict the RNA binding affinity with both mutant and wild-type [50–52].

## Molecular dynamics simulation

The stability and dynamic behavior of the modeled structure were investigated based upon the Desmond with 2005 force field i.e., (OPLS) Optimized Potentials for Liquid Simulations [53]. To examine the multiple conformations and interactions of the ligands within the binding pocket of YBMV CP, we used Desmond molecular dynamics simulations. TIP3P was used as a solvent in an orthorhombic box and the protein was neutralized by introducing counter ions and 0.15M of salt concentration to neutralize the protein [54]. The distance between the box wall and the protein was fixed at 10 Å microns to avoid steric interference with the protein periodic images. Energy minimization was performed using a hybrid approach of steepest descent and limited memory Broyden–Fletcher–Goldfarb–Shanno (LBFGS) algorithms with a maximum of 5000 steps until a gradient threshold was reached up to 25 kcal/mol [55]. The NVT (number of particles, Volume, and Temperature) ensemble conditions were kept the same for 1ns to get simulation data for post-simulation analysis [56]. Nose-Hoover thermo-stats were used to keep the temperature at 300 degrees Celsius throughout the simulation, and the Martyna-Tobias-Klein barostat method was employed to keep the pressure at a consistent level throughout. A multi-step RESPA integrator technique was utilized to investigate the equation of motion in dynamics [57]. The completed equilibrated system was used to run a 50 ns MD simulation, which was examined using Desmond's event analysis module. The docking complexes were employed as starting structures for a 50 ns MD simulation, which was performed on them. Structures were imported into Desmond's set-up wizard and soaked with TIP3P inside an orthorhombic box, like a box with a top and bottom. All other procedures were carried out in the same manner as described in the free form of the YBMV CP protein simulation.

## Results

### Sequence analysis of YBMV CP

Yam bean mosaic virus (YBMV) coat protein (CP) sequence was retrieved from the NCBI protein database (YP_004940328) consisting of 275aa. The domain prediction using SMART, Pfam, and CDD (conserved domain database) revealed a common conserved region of the coat protein sequence of the YBMV *i.e.* (41-270aa) 230aa. This conserved region is utilized further for the structural analysis. The primary sequence analysis of YBMV CP (230aa long) illustrated that the protein is acidic (an isoelectric point is 6.26) in nature and has a molecular mass of 26.41kDa. The aliphatic index is 75.13 which indicate the protein's stability over a wide range of temperature. It is widely recognized that a protein with an instability index less than 40 is

considered to be stable, while if the value ranges above 40 it shows the protein to be unstable [58]. The coat protein (CP) of YBMV is stable as its instability index is 37.02 (*i.e.*, <40). The protein contains 31 negatively charged and 29 positively charged amino acids. YBMV CP has a Grand Average Hydropathicity GRAVY value of -0.576, indicating that it may have better contact with water. The phosphorylation sites of YBMV CP protein were determined using Net-Phos 2.0 [35] which indicates 5 serine, 5 threonine, and 2 tyrosine sites for phosphorylation.

## Secondary structure analysis and 3D prediction

The secondary sequence analysis reveals that, sequence annotation is influenced by expected features like TM and secondary structure mappings. PSIPRED performs the secondary structure analysis where the coils/turns get dominated over strands and helices along with the secondary structure elements. The secondary structure map discovers the amino acid residues which have the propensity to be a beta-sheet or an alpha-helix are present in the proteins derived from the YBMV CP sequence. In addition, sequence characteristics such as the DomPred boundary, disordered area, disordered protein binding sites, and DomSSEA boundary are discovered and annotated in the sequence (**S1 Fig**)**.** The protein sequence of YBMV CP shows the presence of the transmembrane helix region starting from 76aa up to 91aa. It also depicts the disordered protein binding area, which spans 215 to 233 amino acids. Sequence features like cytoplasmic region, pore-lining helix and extracellular region are annotated and predicted. The contact maps of YBMV CP revealed the amino acid residue-residue interactions (**S2 Fig**).

## Phylogenetic analysis

To understand the molecular evolution of YBMV from *Pachyrhizus erosus*, phylogenetic analysis was performed on 26 different coat proteins of Potyviruses belonging to the family Potyviridae (see **S1 Table**)**.** The Neighbor-Joining method with 1000 iterations was used to create the 2D phylogenetic tree and branch length 4.229 has been displayed using MEGA version 6 (**Fig 1**). The Poisson correction technique was used to compute the evolutionary distances. The phylogenetic cladogram represents altogether five clusters and three major clusters formation. On the other hand, it was found that the CP sequence of YBMV shows closely related to CP of Soybean mosaic virus, Bean common mosaic virus and Watermelon mosaic virus. Clusters 3, 4 and 5 show highly dissimilar clades *i.e.*, less sequence similarity to YBMV coat protein. Multiple sequence alignment was performed using Clustal W with the coat protein sequences of Yam bean mosaic virus, Dasheen mosaic virus, Leek yellow stripe potyvirus, Chilli veinal mottle virus, Watermelon mosaic virus, Beet mosaic virus, Onion yellow dwarf virus, Bean common mosaic virus, Cocksfoot streak virus, Bean yellow mosaic virus, Cowpea aphid-borne mosaic virus, Johnson grass mosaic virus, Soybean Mosaic virus, Lily mottle virus, Maize dwarf mosaic virus, Papaya ringspot virus, Peanut mottle virus, Yam mosaic virus, Japanese yam mosaic virus, Zucchini yellow mosaic virus, Turnip mosaic virus, Wild potato mosaic virus, Sugarcane mosaic virus, Sweet potato feathery mottle virus, Sorghum mosaic virus. Multiple sequence alignment of YMBV CP (coat protein) sequence with other CP viruses from the potyvirus family (**S3 Fig**).

## 3D modeling

The search for the templates for YBMV CP in PDB using Delta Blastp explained the crystal structure of Chain A, Watermelon Mosaic Virus (WMV) coat protein (PDB ID: 5ODV), Chain: A [48] shared the highest homology with an identity with coat protein (Bean common mosaic virus) of 80.44% and Query coverage 97%; Coat protein (Peanut stripe virus) identity 79.57 and Query coverage of 97%. The target-template alignment was conducted using the ESPript 3.0 programme as the comparative modelling hinges on a sequence alignment

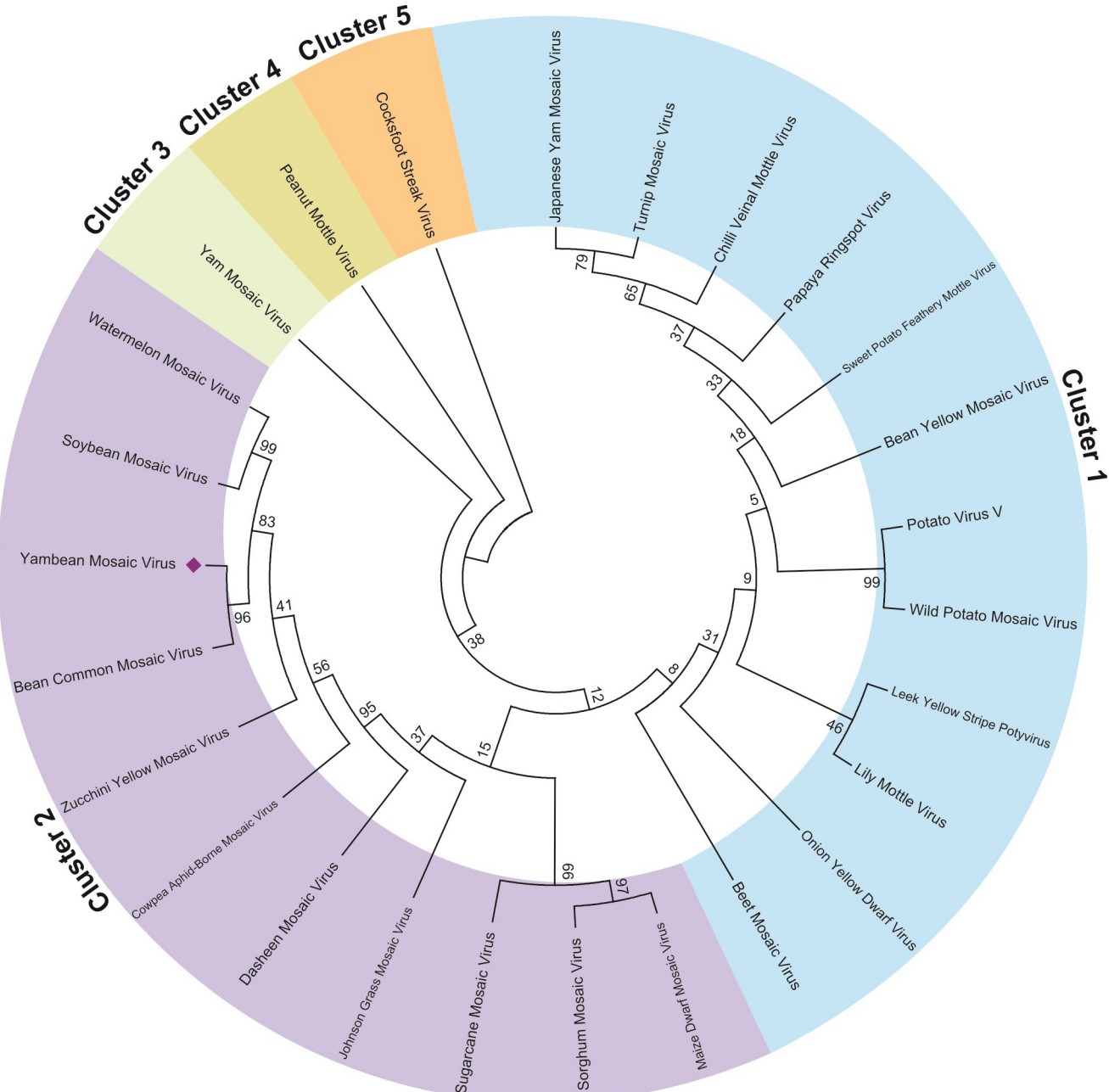

**Fig 1. The evolutionary relationship of taxa of YBMV CP and other closely related organisms using the NJ method.** The bootstrap test (1000 repetitions) showed the percentage of replicate trees where the related taxa were grouped. The analysis was conducted using MEGA version 6.0 software.

between the template and the target sequence with an experimentally determined structure (**Fig 2**). Based on the target-template alignment, the protein model was constructed by the SWISS-MODEL web server.

## Structure validation

The protein structure of the YBMV coat protein comprises two β-strands (one parallel and one anti-parallel) and eight α-helices. The outer surface of the protein is surrounded by the α-

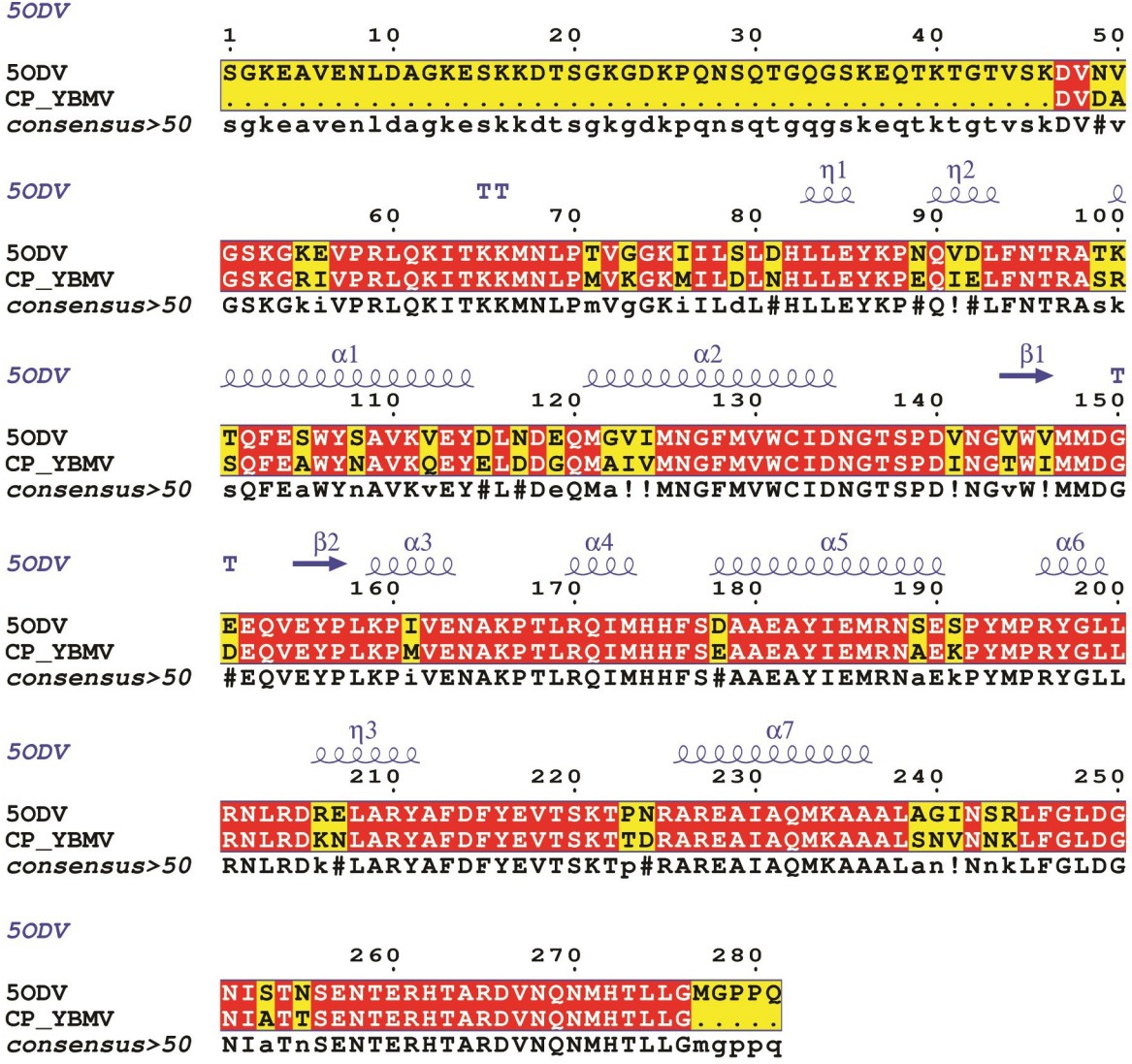

**Fig 2. Target-template alignment of YMBV CP (coat protein) sequence with 5ODV using ESPript 3.0 software.**

helices and the inner core of the model is built of β-sheets (**Fig 3A and 3B**). The dihedral angles (Φ/Ψ) of the YBMV coat protein's accuracy were measured by using the Ramachandran plot in Procheck program (**Fig 4A**). Secondary structure analysis was performed using the PROCHECK analysis tool thus representing the alpha helices and beta sheets in a distinct manner. The 3D model represented a good percentage of residues in favored regions (86.2%), 13.5% residues in additional allowed regions, 0% residues in disallowed regions and 0.5% residues in allowed regions. ERRAT is used in the statistical calculation of the interactions between different atoms types, it also reveals a good validation score is 90%. ERRAT suggests the reliability of the initial model. The ProSA software aims to obtain the proposed model's energy profile as well as the z-score value, which further helps to measure the energy of residue relationship. ProSA evaluated a z-score rating value *i.e.*, as depicted in **Fig 4B**. Protein topology map, constructed using Procheck, with beta-strands as a pink-colored arrow and the alpha-helices as red cylinders (**S4 Fig**). The details of the protein structure validation are summarized in **Table 1**.

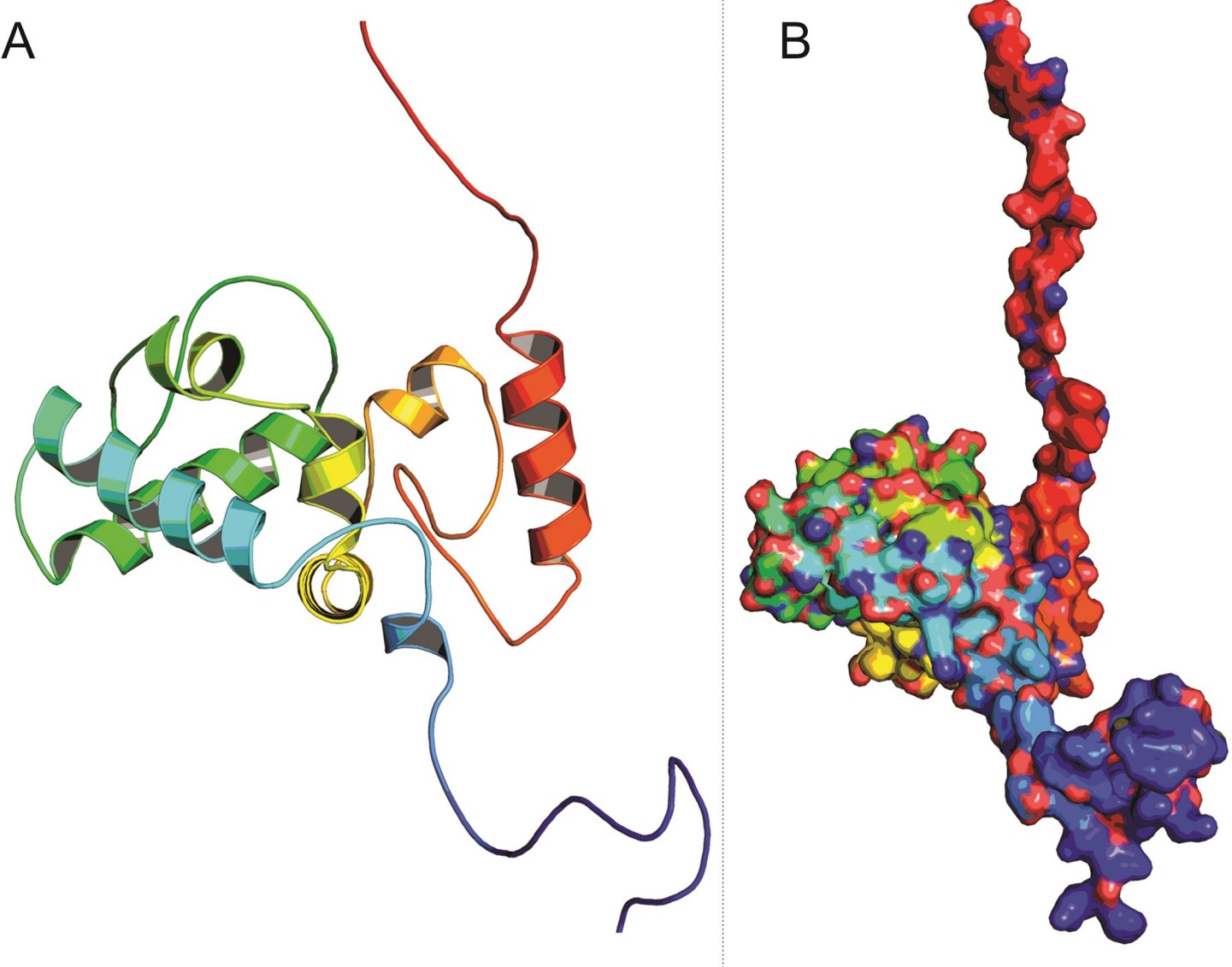

**Fig 3.** (**A**) The three-dimensional structure of YBMV CP N-terminal region (red); C-terminal region (blue). (**B**) The protein is visualized as a solid ribbon representing β-sheets, α-helices, and coils.

## Oligomeric assembly of the conserved fold of YBMV coat proteins

It has been observed that 24 CPs monomeric sub-units form a conserved fold. As a result, we produced its oligomeric form based on the template in our research. The method [53] is based on supervised machine learning and Supports Vector Machines, which integrate interface conservation with other template characteristics structural clustering to produce a quaternary structure quality evaluation. The QSQE score is a number between 0 and 1 that indicates the predicted accuracy of interchain interactions for a model generated with a certain alignment and template. The greater numbers imply more reliance. This is added to the GMQE score, which measures the correctness of the resultant model's tertiary structure.

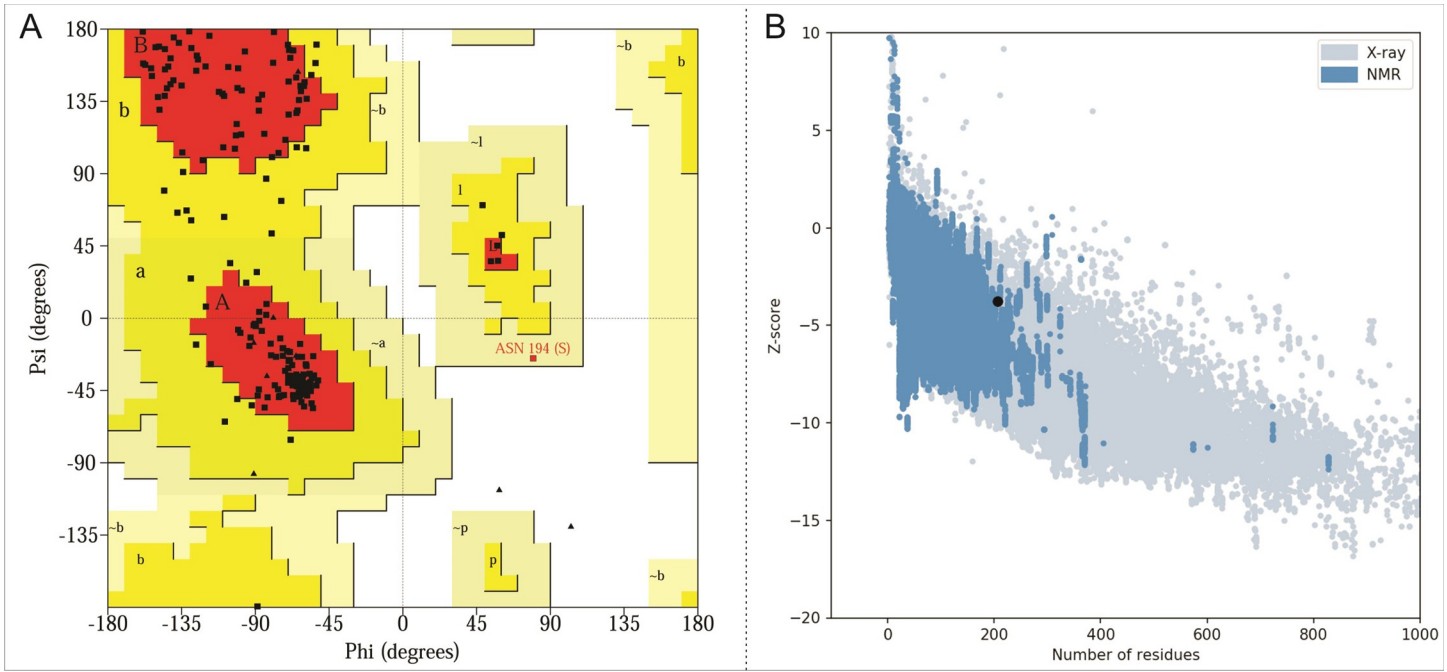

**Fig 4. (A)** Ramachandran plot of the YBMV CP model. The plot was created using the PROCHECK program. **(B)** Protein Structure Analysis (ProSA) of model YBMV CP. The overall quality of the YBMV CP model shows the z-score value of -3.75 (Native conformation to its template).

## RNA binding sites

Interactions between the CP and the viral RNA, in particular, must be tightly controlled for the RNA to be correctly assigned to each of its functions throughout the infection process. In the case of YBMV CP, each subunit covers nucleotides of ssRNA (**Fig 5A–5C**). The presence of a conserved binding site is suggested by analyzing the seven amino acid residues present in the atomic structure of the YBMV CP. The key residues interacting with wild type RNA binding sites are LYS 188, VAL 172, GLU 171, ASN 197, ARG 124 and THR 122 respectively (**Fig 6A**).

**Table 1. Model validation scores of yam bean mosaic virus coat protein (YBMV CP).**

| Model Validation Tools | Model validation reports | Scores |
|---|---|---|
| **PROCHECK** | Most favored region (%) | 86.2% |
| | Additional allowed region (%) | 13.3% |
| | Generously allowed region (%) | 0.5% |
| | Disallowed region (%) | 0.0% |
| | Overall G-factor | 0.18 |
| **Verify3D** | Averaged 3D-1D Score > 0.2 | 61.35% |
| **ERRAT** | Overall quality (%) | 90.00 |
| **ProSA** | Z-score | -3.75 |
| **ProQ** | Predicted LG score | 8.547 |
| | Predicted Max Sub | -0.257 |
| **Molprobity** | Ramachandran outliners (%) | 0.00% |
| | Poor rotamers (%) | 0.00% |
| | Bad bonds (%) | 0.29% |
| | Bad angles (%) | 1.34% |

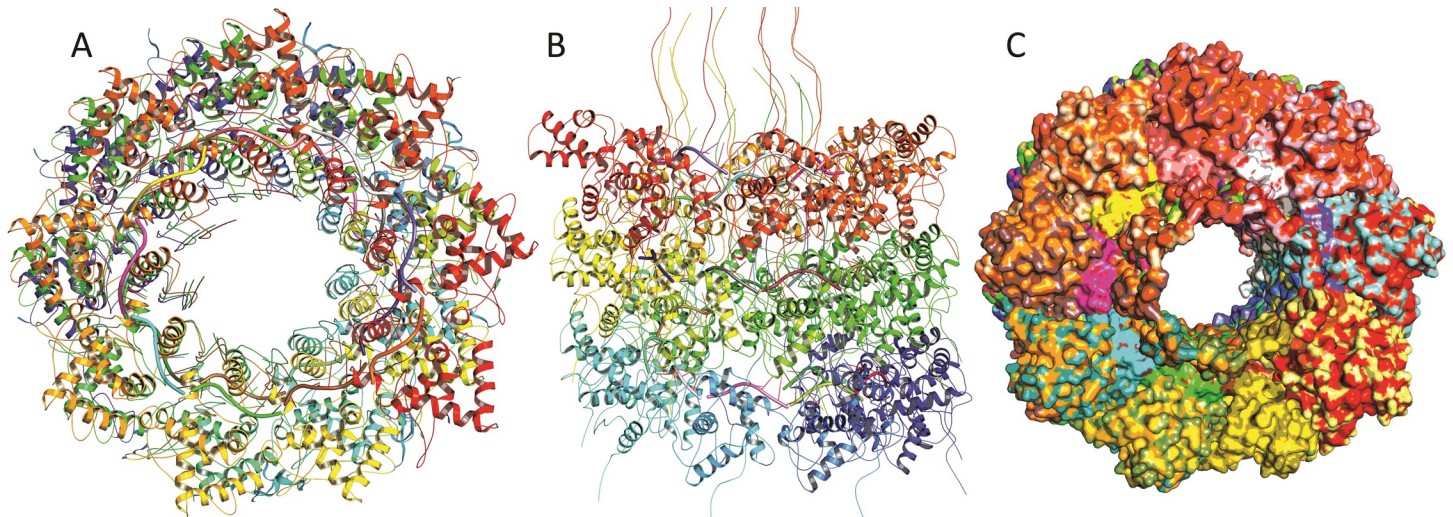

**Fig 5.** Oligomeric assembly of 24 identical subunits of CP with RNA (A) top, (B) front and (C) surface representation).

The key residues interacting with mutant type RNA binding sites are LYS 188, SER 92, THR 91, PRO 93, ASP 168, TYR 170, ALA 142, LEU 123 and THR 122 respectively (**Fig 6B**). The inter-molecular contacts of YBMV CP with RNA were also determined using SWISS-MODEL shown in **Table 2**.

A total of 132 structures in 11 cluster(s), which represents 66% of the water-refined models HADDOCK generated. The wild type HADDOCK score (-68.7), Van der Waals energy (-38.6), Electrostatic energy (-218.7) respectively. The total of 160 structures in 8 cluster(s), which represents 80% of the water-refined models HADDOCK generated. The mutant type HADDOCK score (-61.9), Van der Waals energy (-40.6), Electrostatic energy (-166.2) respectively.

## Molecular dynamics simulation of YBMV CP

Several dynamic stability parameters, including the RMSD, Cα-RMSF, Rg of intermolecular H-bonds, were examined with simulation time after the protein and complex of YBMV CP were obtained. RMSD evaluated the initial structure by MD simulation for 50 ns to investigate the effective confirmation of hits, protein stability, and RNA-binding recognition. By analyzing the deviation caused during the simulation, the Root-mean-square deviation (RMSD) was able to estimate the protein's stability in comparison to its initial structure. The RMSD value shows that, the RNA complex has a more stable shape. The average RMSD of the complex and 3D model of YBMV CP was computed 50 ns after the start of the simulation. RMSD for complex 30 ns to 50 ns stable within 4Å while apo protein deviated throughout the simulation from 6–7.5 (**Fig 7A**). YBMV CP was produced in both the apo and holo forms, and the C-alpha atom's RMSF profile for each residue was computed to determine the residue-wise mobility during the simulation. The YBMV CP-RNA system showed more flexibility as compared to the YBMV CP. The complex's C-alpha Root mean square fluctuations (RMSFs) demonstrated that high fluctuation and confirmed the RMSD results. The mean variation plot was used to assess the compounds average RMSF value. The RMSF for the protein complex C-alpha displays less variation than the RMSF for the apo protein (**Fig 7B**). From the examination of the radius of gyration (Rg), it is clear that the compactness of the complex protein was stabilized between 17.0–17.5 Å, but the compactness of the apo protein was disturbed

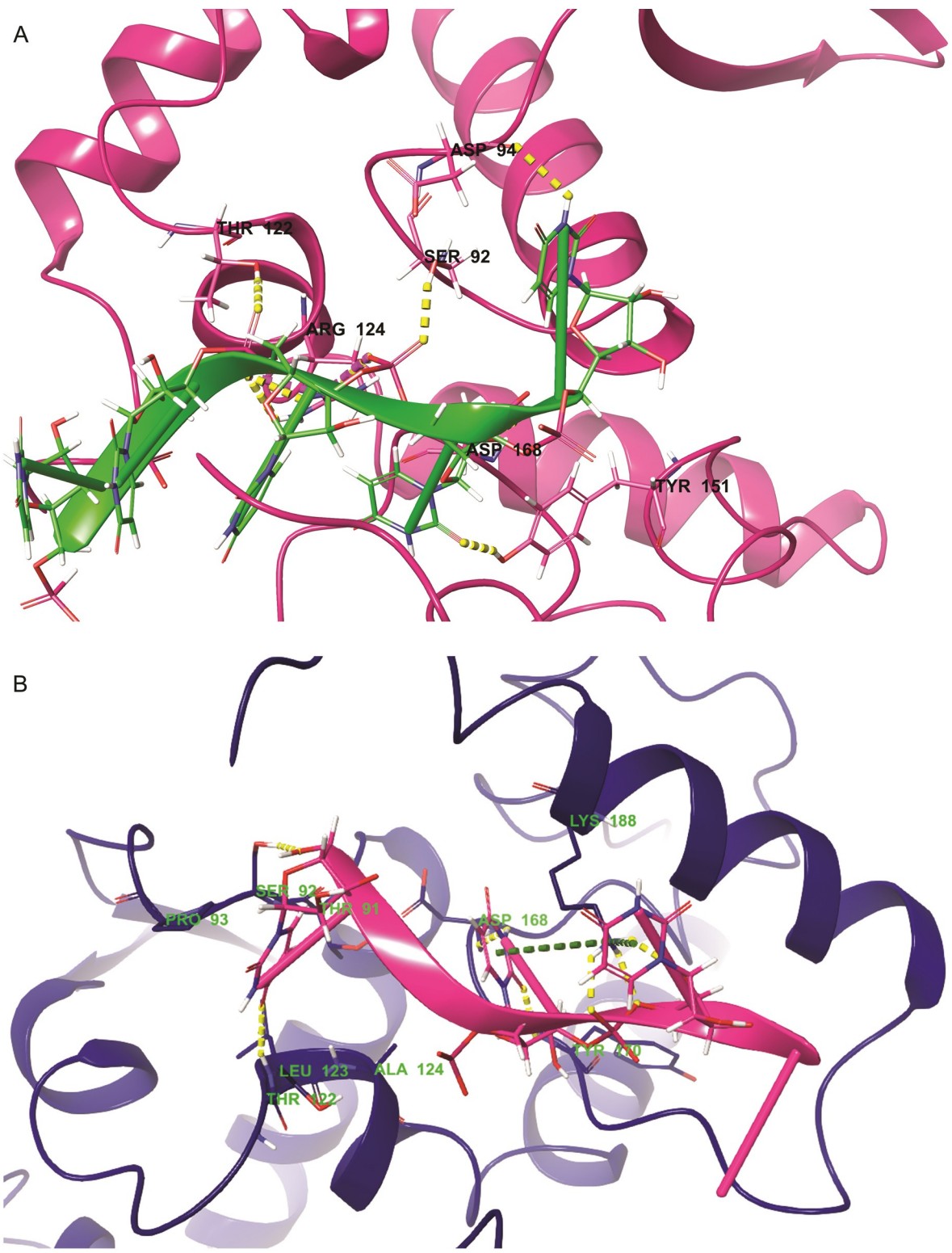

**Fig 6. Interaction analysis of YBMV CP with RNA binding reorganization.** (A) Wild type; (B) R124A mutant.

throughout the simulation. Protein and RNA complexes' compactness and dimensions are determined by it, and this may explain why they interact well. Protein and RNA complexes

**Table 2. Intermolecular contacts YBMV CP with RNA.**

| Pairs | Distance | Type | Category |
|---|---|---|---|
| A:ARG124:NH2—B:U3:OP2 | 2.17 | Hydrogen Bond;Electrostatic | Salt Bridge |
| A:ARG124:NH2—B:U4:OP2 | 3.57 | Hydrogen Bond;Electrostatic | Salt Bridge |
| A:ARG155:NH2—B:U5:OP1 | 3.04 | Hydrogen Bond;Electrostatic | Salt Bridge |
| A:ARG124:NH1—B:U2:OP1 | 5.55 | Electrostatic | Attractive Charge |
| A:ARG124:NH1—B:U3:OP1 | 5.59 | Electrostatic | Attractive Charge |
| A:SER92:OG—B:U4:OP1 | 2.04 | Hydrogen Bond | Conventional Hydrogen Bond |
| A:THR122:OG1—B:U3:OP1 | 3.08 | Hydrogen Bond | Conventional Hydrogen Bond |
| A:TYR151:OH—B:U4:O2 | 2.65 | Hydrogen Bond | Conventional Hydrogen Bond |
| B:U5:H3—A:ASP94:O | 2.43 | Hydrogen Bond | Conventional Hydrogen Bond |
| B:U3—A:LEU192 | 5.26 | Hydrophobic | Pi-Alkyl |
| B:U4—A:LYS188 | 5.23 | Hydrophobic | Pi-Alkyl |

compactness and dimensions are determined by it, and this may explain why they interact well (**Fig 7C**). H-bonds are the most essential factors for biomolecular complexes' stability. It was determined that the number of discrete H-bonds formed between certain amino acid residues of the YBMV core protein and RNA atoms was computed using the Desmond with Optimized Potentials for Liquid Simulations (OPLS) 2005 force field. We computed the intermolecular H-bonds generated between YBMV CP and the RNA-Complex as a function of the simulation timeframe. The H-bond study revealed that RNA created 8–12 hydrogen bonds with protein during the replication process (**Fig 7D**). Overall, taking into consideration of dynamics stability, H-bond analyses and the pre and post-MD RNA binding views, it may be concluded that was more stable.

## Wild type

The ARG side chain has four structure-based donors that can all hydrogen-bond to a single H-bond acceptor.

Asp 168 showed more than 95% direct interaction and formed water mediated interaction more than 35% with the RNA during simulation. Arg124 established 3 direct interactions with the RNA and stabilized more than 60% of the molecular dynamics simulation (**Fig 8A**). **Fig 8B** represent the different bond interaction of the amino acid resides with RNA. Arg124 interacted with RNA by forming mostly H-bond and water bridge. AGR 155 communicated with RNA through H-bond, salt bridge and water bridge. The bottom panel shows which residues interact with the ligand in each trajectory frame. Some residues make more than one specific contact with the ligand, which is represented by a darker shade of orange, according to the scale to the right of the plot (**Fig 8C**).

## Mutant type MD simulation

We have used different stability predictors to assess the effect of mutation of the key residues. The RMSD value shows that the R124A and complex has a more stable during 50 ns MD simulation. RMSD for complex 15 ns to 45 ns stable within 4Å while apo protein deviated throughout the simulation (**Fig 9A**). The complex's C-alpha Root mean square fluctuations (RMSFs) demonstrated that high fluctuation and confirmed the RMSD results. The RMSF for the R124A and complex C-alpha displays less variation than the RMSF for the apo protein (**Fig 9B**). The examination of the radius of gyration (Rg), it is clear that the compactness of the complex protein was stabilized between 17.0–15.5 Å, but the

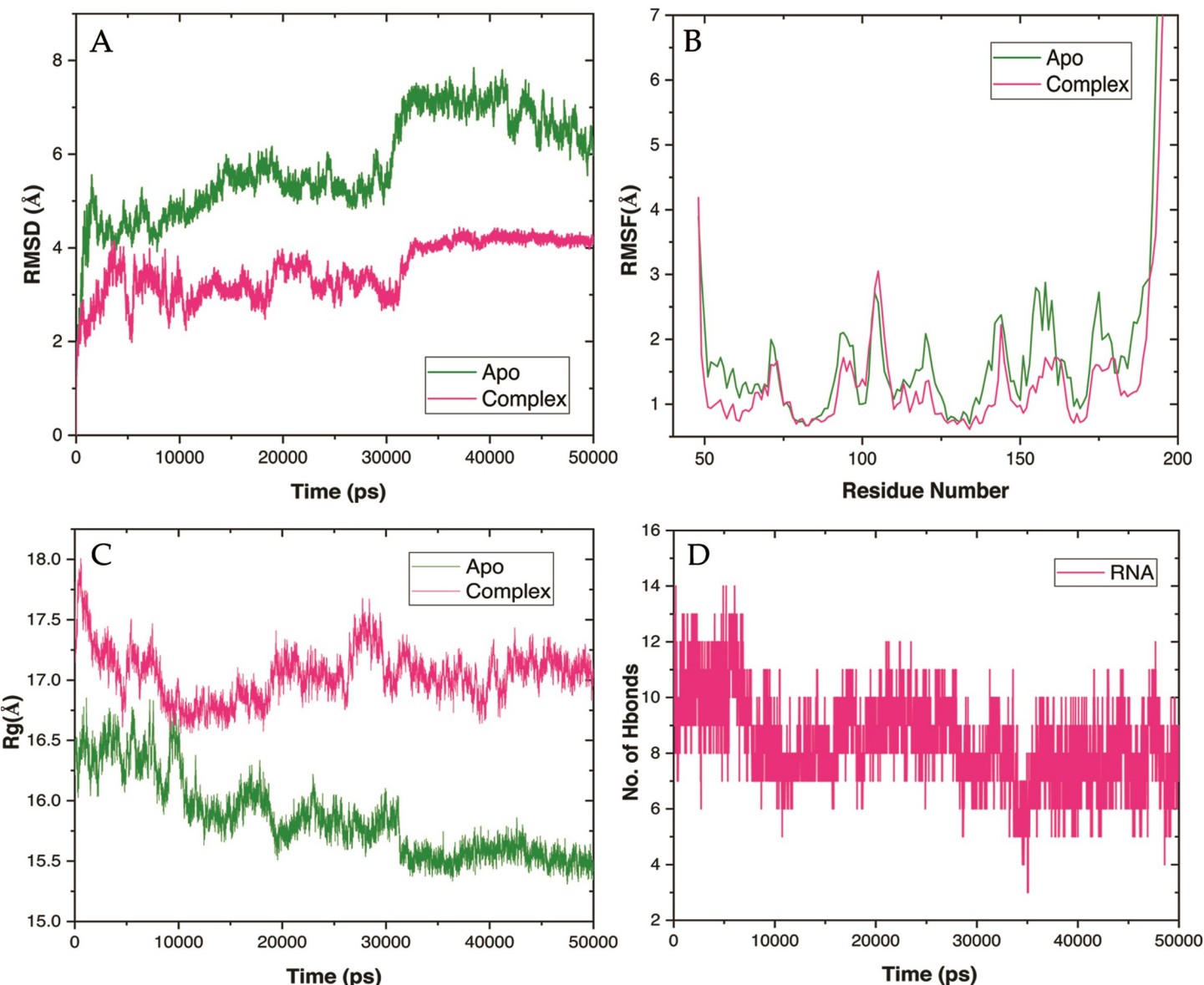

**Fig 7. Conformational stability of YBMV CP of apo and complex through MD simulation.** (**A**) Backbone RMSD (**B**) C-alpha RMSF YBMV CP (**C**) Radiation of gyration for YBMV CP; (**D**) Hydrogen bond analysis over 50 ns period concerning the index of the residue.

compactness of the R124A protein was disturbed throughout the simulation (**Fig 9C**). H-bonds are the most essential factors for biomolecular complexes' stability. It was determined that the number of discrete H-bonds formed between certain amino acid residues of the R124A and R124A Complex was computed using the Desmond with Optimized Potentials for Liquid Simulations (OPLS) 2005 force field. We computed the intermolecular H-bonds generated between R124A and R124A Complex as a function of the simulation timeframe. The H-bond study revealed that created 2–14 hydrogen bonds with protein during the replication process (**Fig 9D**). Overall, taking into consideration of dynamics stability, H-bond analyses and the pre and post-MD RNA binding views, it may be concluded that was more stable.

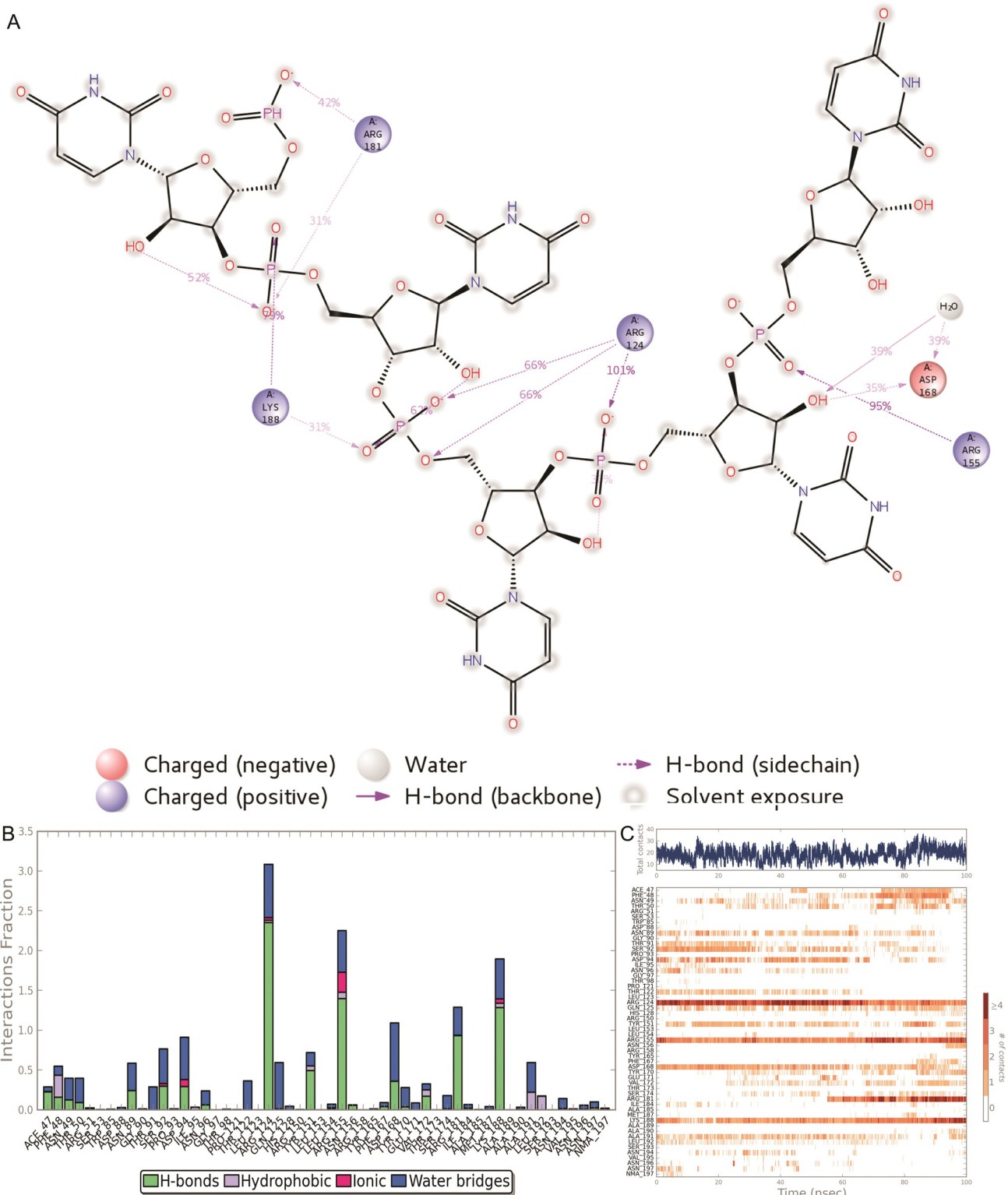

**Fig 8. The H-bond occupancy of wild type YBMV CP with RNA throughout the trajectory.** (**A**) 2D interaction of RNA with YBMV CP during simulation (**B**) the amino acid residues involved in the interactions (**C**) the overall interaction of the protein with RNA.

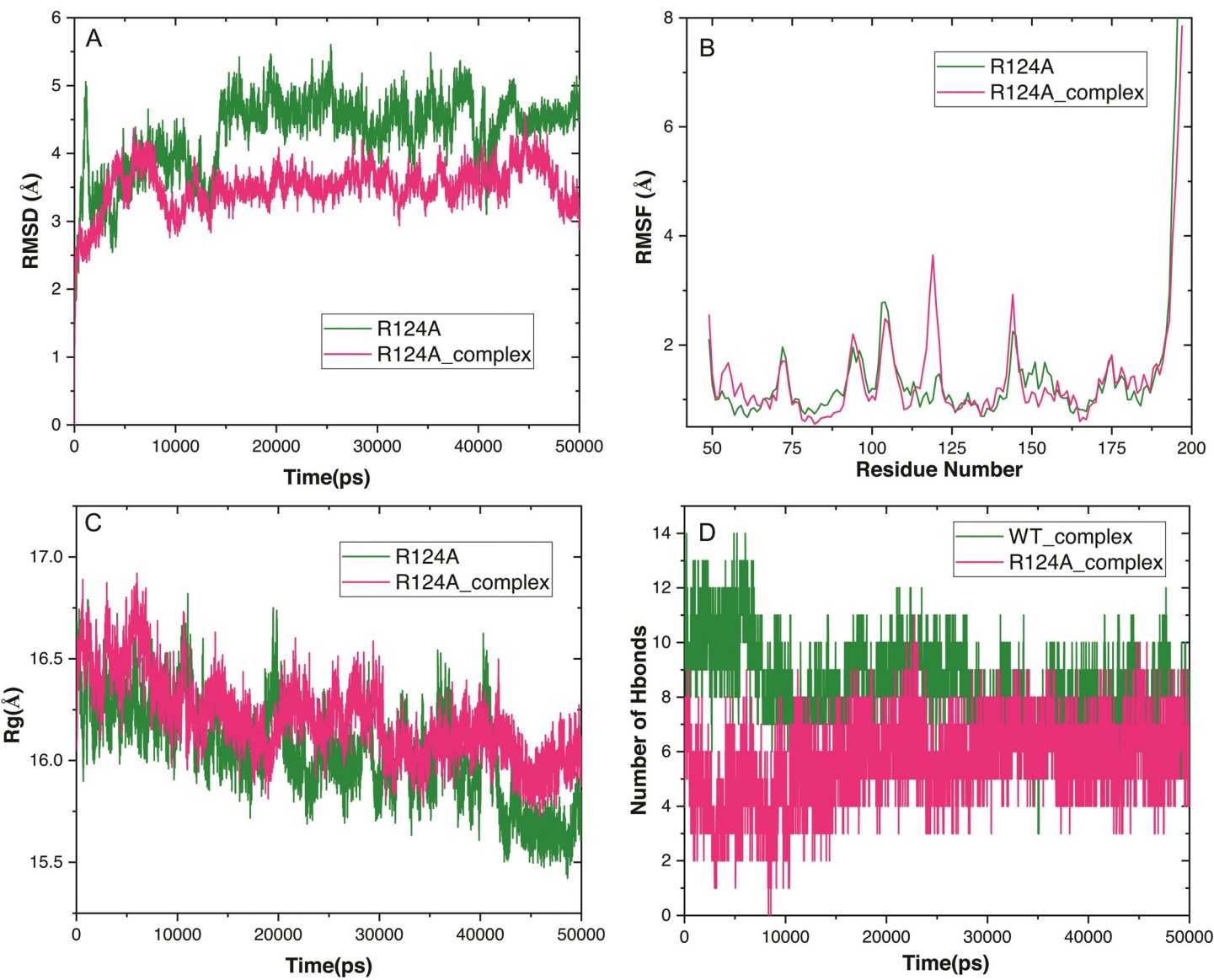

**Fig 9. Conformational stability of YBMV CP of R124A_Apo and R124_Complex through MD simulation. (A)** Backbone RMSD (**B**) C-alpha RMSF YBMV CP (**C**) Radiation of gyration for YBMV CP; (**D**) comparison of Hydrogen bond analysis between wild type and R124A mutant over 50 ns period concerning the index of the residue.

## Mutant type

The **Fig 10A** disclosed that Ala124, Leu123, Pro121, Tyr170, Lys180 and Thr176 established direct interaction with the RNA and stabilized more than 30% throughout simulation. Additionally, it has been noticed that ARG155 communicated with RNA through $H_2O$ molecule. Furthermore, distribution of interaction of each amino acid residues has been displaced in **Fig 10B**. A timeline representation of the interactions and contacts H-bonds, Hydrophobic, Ionic, Water bridges. The bottom panel shows which residues interact with the RNA in each trajectory frame. Some residues make more than one specific contact with the ligand, which is represented by a darker shade of orange, according to the scale to the right of the plot (**Fig 10C**).

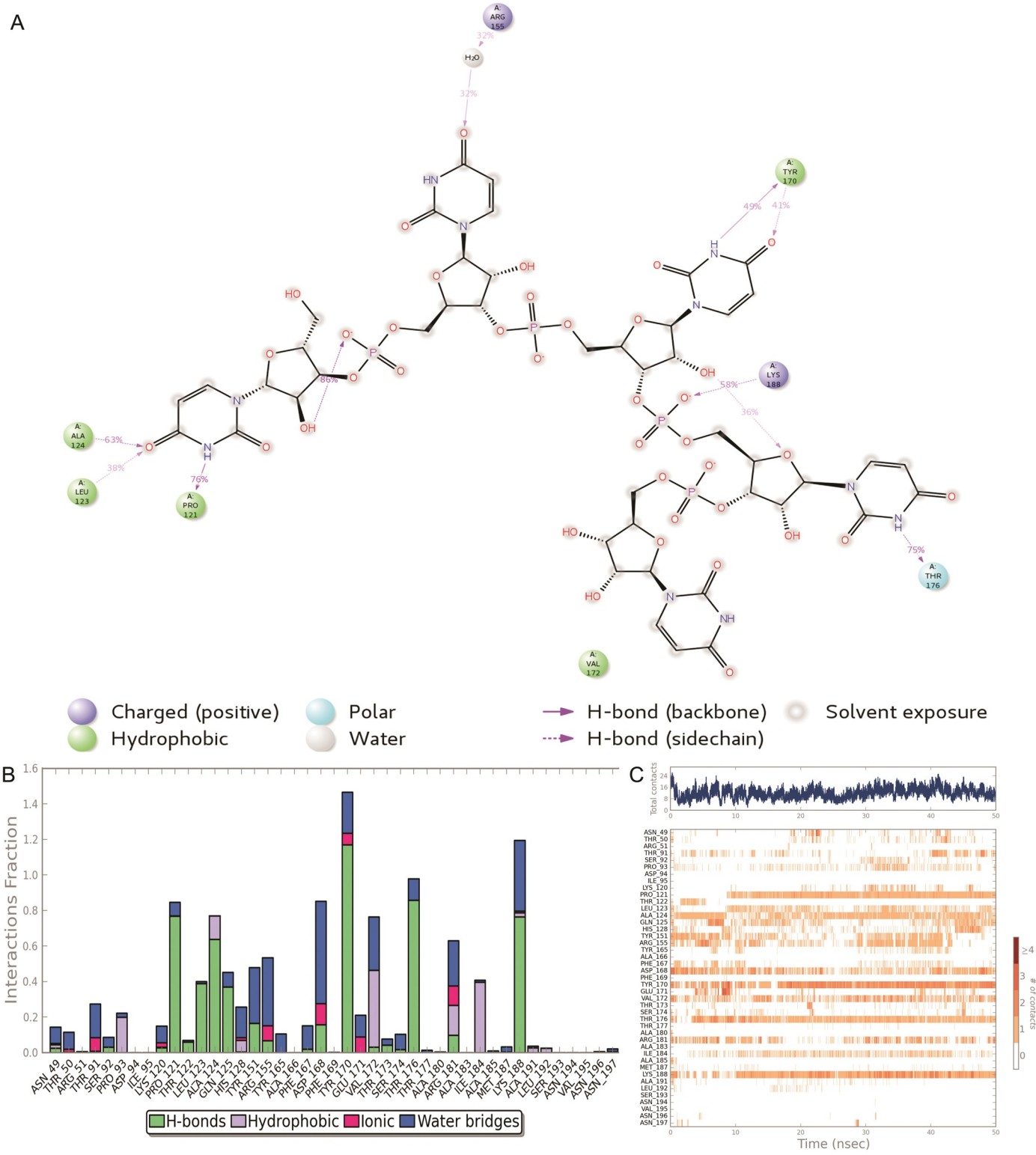

**Fig 10. The H-bond occupancy of mutant type YBMV CP with RNA throughout the trajectory. (A)** 2D interaction of RNA with YBMV CP during simulation **(B)** the amino acid residues involved in the interactions **(C)** the overall interaction of the protein with RNA.

## Discussion

Among legumes, yam beans (*Pachyrhizus erosus*) have the greatest adaptability and production potential [10]. Potyvirus is considered to be the largest and economically most important plant virus. The potyvirus consists of 28 plant virus families and groups. In 1974 total of 1112 species of 369 genera in 53 plant families has been reported [30]. One-quarter of known plant RNA viruses belong to the potyvirus group family potyviridae [59, 60]. Presently there are 2,026 no. of plant species and 566 no. of genera along with 81 botanical families that are globally distributed, which are subjected to potyviruses [61]. More than 200 species of aphids are responsible for the transmission of the potyvirus occurring in a non-persistent way [59, 62]. As a result, potyviruses have a unique capacity to adapt to novel habitats, hosts, and vectors, according to Nigam and co-authors. The investigation of polyprotein-wide and genome-wide variation indicated that the potyviral genome possesses hypervariable locations that preferentially collect nucleotide substitutions and sites under positive selection, encouraging protein disorder and perhaps determining host adaptability [63]. The potyviruses are found in numerous pathogens or strains which vary in biological properties *i.e* pathogenicity [64]. According to Fuentes et al., (2012), a new virus known as Yam Bean Mosaic Virus (YBMV) was discovered and named, and it belongs to the potyviridae family. The virus was first reported by [19], and by default, the name was given as BCMV-IYbn. Latter in 2012 after further study on the virus Fuentes discovered the virus in the yam bean crop and named it Yam Bean Mosaic Virus (YBMV). There is no clear identity, of the YBMV CP core amino acid sequence with the corresponding sequences of members of the genus Potyvirus. It is necessary to achieve a better knowledge of the coat protein structure of YBMV which may help to understand the virus's multiple functions. The N-terminal portions of potyvirus CP sequences are the most diverse, whereas the C-terminal half is very conserved [62]. Thus, the capsid protein and 3' NCR sequences are the most essential parts of the potyviral genome for taxonomic study, and the 3'-terminal area may be utilized to identify virus strains from other potyviruses, according to all molecular data [62, 65].

According to Shukla and Ward (1989), the nucleotide sequence of the CP gene among strains of a certain potyvirus species was more than 90% similar [31], whereas the degree of similarity in the nucleotide sequences of the 3' UTR across strains of specific potyvirus species ranged from 83–99 percent, whereas different virus species had identities ranging from 39–53 percent, according to Frenkel et al (1989) [66]. Adams et al 2005 looked at several potyvirus sequences and determined that the greatest possible CP nucleotide sequence identity in the same potyvirus species was 76–77 percent, while the highest possible CP amino acid sequence identity was 82 percent [67].

The virus is described as a new, positive-sense, single-stranded RNA virus enveloped in a novel way [19]. Instead of the full-length sequence, only the domain region was taken for the structural analysis for accurate optimization. The viral sequence is depicted as a novel enveloped, negative-sense as well as single-stranded RNA virus whose strain belongs to the country named Peru. The virus comes under the potyvirus group, a family named potyviridae. The overall small RNA size distribution revealed a significant preponderance of 21-nt RNAs, which is consistent with prior severe potyviral infections [20]. A phylogenetic analysis was carried out representing five clusters from which two are considered to be major clusters. From our study, it was found that the YBMV CP shows high similarity with Bean Common Mosaic Virus (BCMV) and then secondly with Soybean Mosaic Virus and then with Watermelon Mosaic Virus, the remaining cluster illustrates highly dissimilar clades. Currently, the isolate taken for the analysis is from Peru [22] and the respective amino acid sequence is available in NCBI. The primary structure analysis, as well as the sequence annotation, is essential for

converting the sequence data to functional genes which is essential to understanding the functional and biological mechanisms [68]. In addition to direct functional analysis, the precise position of the transmembrane helices (TM) is important for functional annotation [69]. The YBMV contains the TM region starting from 76aa up to 91aa. For understanding, the functional and the biological mechanisms sequence annotation and physiochemical property analysis are essential, which are obtained after converting the sequence data into functional genes [70]. Preliminary attempts were made to obtain structural information on the coat protein of YBMV by using *in-silico* approaches yielding certain results.

It is important to know how proteins work in three dimensions to answer many biological questions. However, the number of genes and genomes that have been sequenced is quickly outpacing the number of experimentally determined structures. In this approach, comparative modeling continues to be a highly effective strategy of choosing. It contributes to the closure of the gap between what we know about the sequence of a protein and how it appears by creating precise and trustworthy protein models [71].

To solve many biological problems there is a need to have good knowledge regarding the three-dimensional protein structure of the virus (YBMV). Nowadays, comparative modeling is performed as it is an increasingly important method to obtain a suitable 3D protein structure [70, 72–75]. Comparative modeling's main benefit is that it aids in filling in gaps between available sequence and structural data by generating a reliable and accurate protein model [71]. The approach and parameters of MD simulation were adapted from our early studies to better comprehend the molecular function of unknown/hypothetical molecules [76–81].

In our work, 3D modeling is performed following the template search for YBMV coat protein (PDB ID: 5ODV, Chain: A) [48], resulting in appropriate 3D models for a wide range of applications. However, the model, structural stability, and protein-ligand interaction studies also provide prior knowledge for treatments creation soon. Molecular dynamics simulation provides a very realistic, safe, and straightforward atomic level technique for understanding protein folding and dynamics over a certain timescale [82].

Using a bioinformatics approach, several computational tools are provided to examine various dynamical behaviours of the protein structure. In this work, we examined our protein and the complex of YBMV CP utilizing several characteristics such as RMSD, CRMSF, and Rg of intermolecular H-bonds. The original structure was analyzed for RMSD using MD simulation for 50ns to study the effective confirmation of hits, protein stability, and RNA-binding recognition. The average root mean square deviation (RMSD) of the complex and three-dimensional model of YBMV CP was calculated 50ns after the simulation. The apo protein's RMSD fluctuated from 6–7.5 throughout the experiment, while the RMSD for the complex remained steady at 4. (**Fig 7A**). Less variation is seen in the RMSF for C-α, a complex of proteins, than in the RMSF for apo C-α. (**Fig 7B**). The radius of gyration (Rg) shows that the compactness of the complex protein was stable between 17.0–17.5, but the compactness of the apo protein was not stable at all (**Fig 7C**). H-bonds are the most important variables in determining the stability of biomolecular complexes. A force field based on the Desmond with Optimized Potentials for Liquid Simulations (OPLS) 2005 force field was used to calculate the number of discrete H-bonds generated between specific amino acid residues of the YBMV core protein and RNA atoms. According to this research, we computed the intermolecular H-bonds produced between YBMV-CP and the complex of RNA molecules as a function of the simulation period. It was discovered through the H-bond study that RNA formed 8–12 hydrogen bonds with protein during the replication process (**Fig 7D**).

The emergence of this virus disease might pose a significant threat to India's newly launched and promising yam bean Mosaic Virus output. Other significant factors to assess the threat of this virus to global yam bean output include the virus's worldwide prevalence,

variability, susceptibility, and influence on the yield of different yam bean species. Our results will also help in the development of novel antiviral medicines that target the RNA binding pocket, which might disrupt the genome packaging and assembly of economically significant plant viruses like YBMV [48]. Overall, we anticipate that our unique insights into RNA interaction mechanism in Yam bean Mosaic Virus (coat protein), as well as the new study on virus particles, will lead to control of potyviral infection in the future.

## Conclusions

Understanding the significance of a viral protein in the infection process involves a detailed three-dimensional structure of the protein. Our study substantiates from the molecular evolutionary analysis that the YBMV CP of yam bean is highly similar to the coat protein of Bean common mosaic virus, Soybean mosaic virus, Watermelon mosaic virus and potyviridae family. We systematically searched the desired YBMV CP sequence for the *in-silico* analysis and homology-driven structure prediction was generated and characterized in yam bean. The emergence of this new viral illness might pose a significant danger to the yam bean production that has only lately begun and appears to be promising. The entire genome of a novel potyvirus could be reconstructed from the siRNA sequences, and it was distinct enough from other sequences to be classified as a new species. Future research should focus on developing precise detection methods and determining the degree of seed transfer so that suitable phytosanitary measures may be developed and implemented to prevent the disease from spreading further. Other critical characteristics needed to assess the threat of this virus to global yam bean output include the virus's worldwide prevalence, variability, susceptibility, and influence on the yield of diverse yam bean species.

## Supporting information

**S1 Fig. The secondary structure map.** The figure represents the secondary structure map where query sequence and coloured residues are presented as per the annotations made by each analysis methods. Map shows helical residues that are coloured in pink, b-strand residues coloured in yellow, putative domain boundaries are indicated in blue, and green bordered box represents disordered protein binding.
(TIF)

**S2 Fig. The contact map.** The figure shows the contact map of YBMV protein model. The map shows a matrix with contacts with the color scale which represents the relationship between the amino acids in the particular map.
(TIF)

**S3 Fig. Multiple sequence alignment.** Multiple sequence alignment of YMBV_CP (coat protein) sequence with other CP viruses from the potyvirus family. The image is generated in Multalin software.
(TIF)

**S4 Fig. Protein topology map.** Protein topology map, constructed using Procheck, with beta-strands as a pink-coloured arrow and the alpha-helices as red cylinders.
(TIF)

**S1 Table. Identification of protein domain.** The full-length sequence acc. no. of the coat protein of potyvirus and identification of protein domain using the conserved domain search.
(DOC)

## Acknowledgments

The authors are thankful to the Director, ICAR-Central Tuber Crops Research Institute, Trivandrum, Kerala, India for kind support to carry out the research work.

## Author Contributions

**Conceptualization:** R. Arutselvan, Kalidas Pati.

**Data curation:** Ajaya Kumar Rout, Budheswar Dehury.

**Formal analysis:** Varsha Acharya, Ajaya Kumar Rout, Budheswar Dehury.

**Investigation:** Varsha Acharya, Kalidas Pati.

**Methodology:** Varsha Acharya.

**Software:** Varsha Acharya, Ajaya Kumar Rout, Budheswar Dehury.

**Supervision:** R. Arutselvan, Kalidas Pati.

**Validation:** Kalidas Pati.

**Writing – original draft:** Varsha Acharya, R. Arutselvan, Ajaya Kumar Rout, Budheswar Dehury.

**Writing – review & editing:** Ajaya Kumar Rout, Budheswar Dehury, V. B. S. Chauhan, M. Nedunchezhiyan.

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
