## [Decision Letter · Decision Letter 0]

16 Mar 2022

PONE-D-22-04164Structural insights into the RNA interaction with Yam bean Mosaic Virus (coat protein) from Pachyrhizus erosus using bioinformatics approachPLOS ONE

Dear Dr. Pati,

Thank you for submitting your manuscript to PLOS ONE. After careful consideration, we feel that it has merit but does not fully meet PLOS ONE’s publication criteria as it currently stands. Therefore, we invite you to submit a revised version of the manuscript that addresses the points raised during the review process.

We look forward to receiving your revised manuscript.

Kind regards,

Rajarshi Gaur

Academic Editor

PLOS ONE

2. We note you have included a table to which you do not refer in the text of your manuscript. Please ensure that you refer to Table 2 in your text; if accepted, production will need this reference to link the reader to the Table.

Reviewers' comments:

Reviewer's Responses to Questions

**Comments to the Author**

1. Is the manuscript technically sound, and do the data support the conclusions?

Reviewer #1: Yes

Reviewer #2: No

2. Has the statistical analysis been performed appropriately and rigorously? 

Reviewer #1: Yes

Reviewer #2: N/A

3. Have the authors made all data underlying the findings in their manuscript fully available?

Reviewer #1: Yes

Reviewer #2: Yes

4. Is the manuscript presented in an intelligible fashion and written in standard English?

Reviewer #1: Yes

Reviewer #2: No

5. Review Comments to the Author

Reviewer #1: Title; Structural insights into the RNA interaction with Yam bean Mosaic Virus (coat protein) from Pachyrhizus erosus using bioinformatics approach.

Comments;

1. Authors have not provided the name of the template used for model generation in the abstract.

2. Authors have suggested to rewrite the results and conclusion part of the abstract based on findings. In addition, the authors have advised adding a brief description of methodology in the abstract.

3. Objective of the study should be clear at the end of the introduction section.

4. Authors have been advised to correct the spelling mistakes throughout the manuscript. For example “formatted”.

5. In the Introduction section the author should refer to the research paper and comment on recent in-silico techniques. It will be good information for the readers. I would like to recommend several papers, among many others, providing further explanation on this topic:PMID: 34717229 PMID: 33465692 PMID: 33749525 PMID: 29380245 PMID: 31933177 PMID: 32353478 PMID: 33993976

6. Provide citations for all the PDB IDs mentioned in the manuscript.

7. Authors should include studies with RNA on the predicted binding site. This could be done by docking and MD simulations.

8. MD simulations should be carried out to identify key structural changes and residues involved in protein-RNA interactions.

9. Clearly mention the aim and objective of the study in the last paragraph of the introduction section.

10. Including limitations of the study in the discussion and conclusions section.

11. Authors have not provided references regarding servers used in the studies.

12. Figure 6 is not visible authors have advised redrawing the figure.

13. Authors have advised to include more rigorous analysis and validate the findings by experimental/computational analyses.

18. Authors have not compared their findings with already present literature based on similar studies.

19. Overall, the manuscript is written in a very confusing and poor way. The writing needs substantial improvement. Need to edit the English language through a native speaker or by other means.

In my view, the results obtained in this study are worthy for publication. The manuscript needs major essential revision before publication. I would like to overview the revised version of the manuscript before it accept for publication.

Reviewer #2: In this manuscript titled “Structural insights into the RNA interaction with Yam bean Mosaic Virus (coat protein) from Pachyrhizus erosus using bioinformatics approach” authors have modeled the 3D structure of CP and merely predicted the amino acid that is involved in interaction with RNA. There is no experimental evidence for their finding. I recommend the authors plan a wet lab experiment to demonstrate the role of those amino acids in RNA binding. For instance, make mutation of those amino acid residues by SDM and test the ability of mutant proteins to bind RNA by EMSA, fluorescent spectroscopy, etc. The manuscript is not written well.

Specific comments:

Page 3, Line 15: “Chenopodium amanticolor, Nicotiana megalosiphon and Nicotiana benthamiana”. Scientific name should be italicized.

Page 3, Line 80: “Sorensen” since there are more than one author in this study, you should mention as Sorensen et al.,

Page 3, Line 86: Similar to previous comment. Change “Damayanti” to ““Damayanti et al”

Page 4, Line 88: “Afterward in……”. Not scientific. Rephrase this sentence.

6. PLOS authors have the option to publish the peer review history of their article (what does this mean?). If published, this will include your full peer review and any attached files.

Reviewer #1: No

Reviewer #2: **Yes: **PRABU GNANASEKARAN

---

## [Author Response · Author response to Decision Letter 0]

20 Apr 2022

Manuscript Number: PONE-D-22-04164 

Point-to-point response to the Reviewers' comments

We thank the editor for the positive evaluation of our paper, and the two reviewers for recognizing the merit of our work and for providing very valuable suggestions. As detailed below, we have addressed the Reviewers’ comments carefully and revised the manuscript accordingly. To facilitate the reviewing process, we have copied the Reviewers’ original comments, which are shown in black (italic font) and our responses are then shown in blue.

Reviewer #1

Comments 1

Authors have not provided the name of the template used for model generation in the abstract.

Authors Response

The template used for the model generation is 5ODV. We have added the template name in the abstract section of the Ms.

Comments 2

Authors have suggested to rewrite the results and conclusion part of the abstract based on findings. In addition, the authors have advised adding a brief description of methodology in the abstract.

Authors Response

As per the suggestion, the results, conclusion and methodology part has been revised accordingly in the abstract section of the Ms. 

Comments 3

Objective of the study should be clear at the end of the introduction section.

Authors Response

Thank you for your response. We have modified the introduction section as desired by you. 

Comments 4

 Authors have been advised to correct the spelling mistakes throughout the manuscript. For example, “formatted”.

Authors Response

We have corrected our spelling mistakes throughout the manuscript.

Comments 5

In the Introduction section the author should refer to the research paper and comment on recent in-silico techniques. It will be good information for the readers. I would like to recommend several papers, among many others, providing further explanation on this topic: PMID: 34717229 PMID: 33465692 PMID: 33749525 PMID: 29380245 PMID: 31933177 PMID: 32353478 PMID: 33993976

Authors Response

Thank you for your valuable comments. These publications are highly useful regarding in silico techniques and we have also cited the same in the Ms accordingly.

Comments 6

Provide citations for all the PDB IDs mentioned in the manuscript.

Authors Response

We have cited all PDB IDs in the Manuscript. 

Comments 7

Authors should include studies with RNA on the predicted binding site. This could be done by docking and MD simulations.

Authors Response

Thank you for your suggestion. We have performed an MD simulation of 50ns using Desmond with Optimized Potentials for Liquid Simulations (OPLS) 2005 force field.

Comments 8

MD simulations should be carried out to identify key structural changes and residues involved in protein-RNA interactions.

Authors Response

As suggested, we performed an MD simulation to analyze the key structural changes and residues involved in protein-RNA interaction in the Ms.

Comments 9

Clearly mention the aim and objective of the study in the last paragraph of the introduction section.

Authors Response

The aim and objective of the study have been rectified and explained properly in the introduction section of the Ms.

Comments 10

Including limitations of the study in the discussion and conclusions section.

Authors Response

We have modified the discussion and conclusion section as per your suggestion.

Comments 11

Authors have not provided references regarding servers used in the studies.

Authors Response

Thank you for your recommendation. We have cited some references in the Manuscript.

Comments 12

Figure 6 is not visible authors have advised redrawing the figure.

Authors Response

We have revised figure 6 as per your suggestion.

Comments 13

Authors have advised to include more rigorous analysis and validate the findings by experimental/computational analyses.

Authors Response

We have performed a molecular dynamics simulation to validate more findings in computational bioinformatics techniques.

Comments 14

Authors have not compared their findings with already present literature based on similar studies.

Authors Response

We have revised the Manuscript with recent literature studies.

Comments 15

Overall, the manuscript is written in a very confusing and poor way. The writing needs substantial improvement. Need to edit the English language through a native speaker or by other means.

Authors Response

We apologize for the inconvenience. We have revised the manuscript accordingly. We have used Grammarly software and corrected the spelling and grammatical errors present in the manuscript and rewritten the portion that was confusing for the reviewer. 

Reviewer #2

Comments 1

There is no experimental evidence for their finding. I recommend the authors plan a wet lab experiment to demonstrate the role of those amino acids in RNA binding. For instance, make mutation of those amino acid residues by SDM and test the ability of mutant proteins to bind RNA by EMSA, fluorescent spectroscopy, etc. 

Authors Response

Thank you very much for your recommendation. We agree with the reviewer on the above remarks that, the findings from the computational study should be validated through an experimental approach. We are a small group of researchers working in computational biology without the support of a wet-lab experimental setup, therefore, we apologize that the suggested experiments by the esteemed reviewer could not be performed at present MS. However, we strongly believe that our computational findings will be helpful for the researchers working in this area to plan and undertake such studies involving site-directed mutagenesis, EMSA and fluorescent spectroscopy studies which will unravel the binding of RNA to the protein. To additionally, support our study, we have performed all-atoms MD simulation for 50 ns to analyze the structural changes and residues involved in protein-RNA interaction and stability of the complex through RMSD, RMSF, Rg and H-bond analysis (Please see in Fig. 7 of the Ms).

Comment 2

The manuscript is not written well.

Authors Response

Thank you for your suggestion. We have revised the manuscript rigorously and rectified the grammatical mistakes throughout the manuscript. 

Specific comments:

Comments 3

Page 3, Line 15: “Chenopodium amanticolor, Nicotiana megalosiphon and Nicotiana benthamiana”. Scientific name should be italicized.

Authors Response

We have corrected the scientific name in italics.

Comments 4

Page 3, Line 80: “Sorensen” since there are more than one author in this study, you should mention as Sorensen et al.,

Authors Response

We have revised the Sorensen as Sorensen et al., in the Ms.

Comments 5

Page 3, Line 86: Similar to previous comment. Change “Damayanti” to ““Damayanti et al”

Authors Response

We have corrected the “Damayanti” to “Damayanti et al” in the introduction section.

Comments 6

Page 4, Line 88: “Afterward in……”. Not scientific. Rephrase this sentence.

Authors Response

As per the reviewer's suggestion, we have rewritten the above line.

---

## [Decision Letter · Decision Letter 1]

29 Apr 2022

PONE-D-22-04164R1Structural insights into the RNA interaction with Yam bean Mosaic Virus (coat protein) from Pachyrhizus erosus using bioinformatics approachPLOS ONE

Dear Dr. Pati,

Thank you for submitting your manuscript to PLOS ONE. After careful consideration, we feel that it has merit but does not fully meet PLOS ONE’s publication criteria as it currently stands. Therefore, we invite you to submit a revised version of the manuscript that addresses the points raised during the review process.

We look forward to receiving your revised manuscript.

Kind regards,

Rajarshi Gaur

Academic Editor

PLOS ONE

Reviewers' comments:

Reviewer's Responses to Questions

**Comments to the Author**

1. If the authors have adequately addressed your comments raised in a previous round of review and you feel that this manuscript is now acceptable for publication, you may indicate that here to bypass the “Comments to the Author” section, enter your conflict of interest statement in the “Confidential to Editor” section, and submit your "Accept" recommendation.

Reviewer #1: All comments have been addressed

Reviewer #2: (No Response)

2. Is the manuscript technically sound, and do the data support the conclusions?

Reviewer #1: Yes

Reviewer #2: Yes

3. Has the statistical analysis been performed appropriately and rigorously? 

Reviewer #1: I Don't Know

Reviewer #2: N/A

4. Have the authors made all data underlying the findings in their manuscript fully available?

Reviewer #1: Yes

Reviewer #2: Yes

5. Is the manuscript presented in an intelligible fashion and written in standard English?

Reviewer #1: Yes

Reviewer #2: Yes

6. Review Comments to the Author

Reviewer #1: The authors have responded to all the concerns meticulously and improved the manuscript accordingly. The revised draft is improved significantly. I do not have further comments.

Reviewer #2: Authors have addressed the comments partially. Still there is no strong evidence for their conclusion. I recommend authors to perform atleast few more computational analyses to support their conclusion.

Specific comments:

Still, there is no wet lab experimental evidence for their finding. But authors have performed MD simulation to analyze the structural changes and residues involved in protein-RNA interaction. However, I recommend the authors to add atleast few more computation analysis, such as,

Generate simulation of mutant protein structure

Determining the wildtype and mutant protein affinity for RNA by Haddock score calculation.

Identify the pocket that binds to RNA, and the amino acid residues involved in the interaction in both wild-type and mutant protein.

To perform these experiments, I suggest authors to cite and follow the following publications.

Gnanasekaran, P., Gupta, N., Ponnusamy, K., and Chakraborty, S. (2021). Betasatellite encoded βC1 protein exhibits novel ATP hydrolysis activity that influences its DNA-binding activity and viral pathogenesis. Journal of Virology, 95 (20): e00475-21.

Gnanasekaran, P., Ponnusamy, K., and Chakraborty, S. (2019) A geminivirus betasatellite encoded βC1 protein interacts with PsbP and subverts PsbP-mediated antiviral defense in plants. Molecular Plant Pathology, 20(7): 943-960.

7. PLOS authors have the option to publish the peer review history of their article (what does this mean?). If published, this will include your full peer review and any attached files.

Reviewer #1: No

Reviewer #2: **Yes: **PRABU GNANASEKARAN

---

## [Author Response · Author response to Decision Letter 1]

4 Jun 2022

Manuscript Number: PONE-D-22-04164R2 

Point-to-point response to the Reviewers' comments

We thank the editor for the positive evaluation of our paper, and the two reviewers for recognizing the merit of our work and for providing very valuable suggestions. As detailed below, we have addressed the Reviewers’ comments carefully and revised the manuscript accordingly. To facilitate the reviewing process, we have copied the Reviewers’ original comments, which are shown in black (italic font) and our responses are then shown in blue.

Reviewer #1

Comments 

The authors have responded to all the concerns meticulously and improved the manuscript accordingly.The revised draft is improved significantly. I do not have further comments.

Authors Response

Thank you for your possetive response.

Reviewer #2

Comments 1

Generate simulation of mutant protein structure

Authors Response

Thank you very much for your recommendation. As per your suggestion, we have generated the mutant structure of R124A which plays important role in the interaction of YBMV_CP with RNA and maintained stable connection throughout the simulation. All the simulation analysis have been included in the MS. 

Comment 2

Determining the wildtype and mutant protein affinity for RNA by Haddock score calculation.

Authors Response

Thank you for your suggestion. We have revised the manuscript accordingly. The HADDOCK score for the wild type and mutant type have been presented in the manuscript.

Comments 3

Identify the pocket that binds to RNA, and the amino acid residues involved in the interaction in both wild-type and mutant protein.

Authors Response

We have identified the pocket that binds to RNA, and the amino acid residues involved in the interaction in both wild-type and mutant type. 

Comments 4

To perform these experiments, I suggest authors to cite and follow the following publications.

Gnanasekaran, P., Gupta, N., Ponnusamy, K., and Chakraborty, S. (2021). Betasatellite encoded βC1 protein exhibitsnovel ATP hydrolysis activity that influences its DNA-binding activity and viral pathogenesis. Journal of Virology, 95(20): e00475-21.

Gnanasekaran, P., Ponnusamy, K., and Chakraborty, S. (2019) A geminivirus betasatellite encoded βC1 proteininteracts with PsbP and subverts PsbP-mediated antiviral defense in plants. Molecular Plant Pathology, 20(7): 943-960.

Authors Response

Thank you for your suggestion. We have cited the two references in the Manuscript.

---

## [Decision Letter · Decision Letter 2]

12 Jun 2022

Structural insights into the RNA interaction with Yam bean Mosaic Virus (coat protein) from Pachyrhizus erosus using bioinformatics approach

PONE-D-22-04164R2

Dear Dr. Pati,

We’re pleased to inform you that your manuscript has been judged scientifically suitable for publication and will be formally accepted for publication once it meets all outstanding technical requirements.

Kind regards,

Rajarshi Gaur

Academic Editor

PLOS ONE

Reviewers' comments:

Reviewer's Responses to Questions

**Comments to the Author**

1. If the authors have adequately addressed your comments raised in a previous round of review and you feel that this manuscript is now acceptable for publication, you may indicate that here to bypass the “Comments to the Author” section, enter your conflict of interest statement in the “Confidential to Editor” section, and submit your "Accept" recommendation.

Reviewer #1: All comments have been addressed

Reviewer #2: All comments have been addressed

2. Is the manuscript technically sound, and do the data support the conclusions?

Reviewer #1: Yes

Reviewer #2: Yes

3. Has the statistical analysis been performed appropriately and rigorously? 

Reviewer #1: Yes

Reviewer #2: Yes

4. Have the authors made all data underlying the findings in their manuscript fully available?

Reviewer #1: Yes

Reviewer #2: Yes

5. Is the manuscript presented in an intelligible fashion and written in standard English?

Reviewer #1: Yes

Reviewer #2: Yes

6. Review Comments to the Author

Reviewer #1: (No Response)

Reviewer #2: Authors have addressed all the comments. My recommendation is to accept the manuscript for publication.

7. PLOS authors have the option to publish the peer review history of their article (what does this mean?). If published, this will include your full peer review and any attached files.

Reviewer #1: No

Reviewer #2: **Yes: **PRABU GNANASEKARAN

---

## [Editor Report · Acceptance letter]

11 Jul 2022

PONE-D-22-04164R2 

Structural insights into the RNA interaction with Yam bean Mosaic Virus (coat protein) from *Pachyrhizus erosus* using bioinformatics approach 

Dear Dr. Pati:

I'm pleased to inform you that your manuscript has been deemed suitable for publication in PLOS ONE. Congratulations! Your manuscript is now with our production department. 

Kind regards, 

on behalf of

Professor Rajarshi Gaur 

Academic Editor

PLOS ONE